# The Epigenetics of Psychosis: A Structured Review with Representative Loci

**DOI:** 10.3390/biomedicines10030561

**Published:** 2022-02-28

**Authors:** Christine L. Miller

**Affiliations:** MillerBio, Baltimore, MD 21239, USA; cmiller@millerbio.com

**Keywords:** schizophrenia, bipolar disorder with psychosis, environmental factors, methylation, gene expression, transcription factor motifs, alternative splicing, principles of evolution

## Abstract

The evidence for an environmental component in chronic psychotic disorders is strong and research on the epigenetic manifestations of these environmental impacts has commenced in earnest. In reviewing this research, the focus is on three genes as models for differential methylation, MCHR1, AKT1 and TDO2, each of which have been investigated for genetic association with psychotic disorders. Environmental factors associated with psychotic disorders, and which interact with these model genes, are explored in depth. The location of transcription factor motifs relative to key methylation sites is evaluated for predicted gene expression results, and for other sites, evidence is presented for methylation directing alternative splicing. Experimental results from key studies show differential methylation: for MCHR1, in psychosis cases versus controls; for AKT1, as a pre-existing methylation pattern influencing brain activation following acute administration of a psychosis-eliciting environmental stimulus; and for TDO2, in a pattern associated with a developmental factor of risk for psychosis, in all cases the predicted expression impact being highly dependent on location. Methylation induced by smoking, a confounding variable, exhibits an intriguing pattern for all three genes. Finally, how differential methylation meshes with Darwinian principles is examined, in particular as it relates to the “flexible stem” theory of evolution.

## 1. Introduction

The realization that chronic psychotic disorders might be controlled by environmental factors was made clear long ago when studies of identical twins revealed a discordance rate for schizophrenia of ~50% [1]. Although only 6% of individuals with schizophrenia have a first-degree family history with the illness [2], the proportion rises to 12–26% if second degree family members are included [3,4]. Having a third degree relative with a psychotic disorder does not significantly raise the risk for schizophrenia above background, as exemplified in the adjustments by Giordano et al. [5] for family history effects in first cousin pairs and is therefore of little relevance. The gap between the ~50% genetic contribution estimate based on the proportion in probands in identical twin studies being concordant for schizophrenia to that derived from the 12–26% of probands having a relevant family history can be viewed as adverse interactions occurring between genes that come in from different sides of the family tree. Indeed, most genetic association studies concerning psychosis now utilize a polygenic approach, creating what is known as the polygenic risk score, or PRS [6], which obviously accommodates potential gene–gene interactions. The remaining 50% of the risk can only be attributable to environmental factors, of which some interact with genes to accentuate risk already posed by those genes, while others may have a more unique impact, interacting with genes that may pose no significant risk absent that environmental factor.

Epigenetic contributions to psychotic disorders invite the comparison of germline variations in genes through natural selection versus environmental conditions which prompt editorial “markup” if you will, of that genetic code. In such a comparison, it is helpful to apply the litmus test of Darwinian principles of evolution to predict the most likely genetic modifier for a particular environmental change, while keeping in mind that the epigenetic mechanism itself is subject to natural selection. Clearly, long-term shifts in the environmental are best dealt with by evolution of allele frequencies in the germline DNA so the organism can more efficiently survive. However, short-term environmental fluctuations that are not fatal for the organism should theoretically not lead to changes in germline allele frequencies because the time frame is too short to exert a differential effect on reproductive success. This appears to be the niche occupied by epigenetics, to allow a response over a time frame when changes in gene frequency will not occur.

The lower limit of the time frame important to invoking epigenetic alterations is an interesting question, as some genes are well-poised to respond immediately to changing biochemical cues in the absence of methylation, as will be explored later for the glucocorticoid response element in the gene for the enzyme tryptophan 2,3-dioxygenase (TDO2). Even epigenetic modifications can come at a cost, however, including a greater risk for point mutations [7]. The principles which determine if the dose and time of exposure to the environmental change is adequate to require epigenetic involvement are not yet clear in the literature. In as little as half an hour of intense exercise, burning over 1674 kJ can result in detectable methylation of CpG sites [8]. However, is that methylation superimposed on a system that has already been primed by prior exercise regimes? Does the short duration of exposure mean the methylation is more readily reversed? These are questions that will not be answered by this review, but are important to keep in mind when thinking about epigenetic events that may be associated with psychosis.

Strictly speaking, epigenetic alterations are those which can survive cellular replication in somatic cells [9], but do not necessarily involve similar DNA modifications in the germline stem-cell DNA [10,11]. Several mechanisms of epigenetic control exist, including methylation of adenine [12], acetylation of histone, and most prominently, methylation of cytosines that lie immediately upstream of a guanine [13], i.e., CpG sites leading to 5-methyl-cytosine (5mC). It is the latter mechanism that will be covered by this review. 

To set the stage for what is to come, it is important to point out a great paradox of methylation, in that 5mC renders the cytosine base more susceptible to transitioning to thymine (T) through spontaneous deamination, the probability of which obviously increases with time [14]. In an environment that promotes consistent epigenetic methylation of a gene, the end result can be the loss of methylation capability at that site, which should be maladaptive given the existing environment that encouraged it in the first place. The explanation for this apparent conflict may lie in the enzymatic reversal of methylation which involve conversion of 5mC to 5-hydroxy-methyl-cytosine (5hmC) as the first step, followed by oxidation intermediates and base excision repair [15]. In such a way, the epigenetic control of the site can be maintained and the replacement of a C by a T minimized. 

Most studies of methylation have not discriminated between 5mC and 5hmC, which has bearing on all of the current research to be reviewed here and is potentially problematic for CpG sites in the gene body but not quite as serious an oversight for the promoter region and the first exon, where the functional outcome is quite similar in terms of expression of the gene. In general, either 5mC or 5hmC in the promoter region and in the first exon repress expression of the gene [16,17,18], whereas 5hmC in the gene body is generally permissive for expression in differentiated cells [19]. 

Publications on genome-wide differential methylation or expression are a great resource for important data on genes relevant to psychosis, for example Montano et al. and Gandal et al. [20,21] and will be discussed extensively in the following sections. Although some controversy exists as to whether the onset of psychosis involves changes in global levels of methylation, with Kebir et al. [22] reporting that it does not, while Tomassi and Tosato [23] contend that global hypomethylation is observed, and Hannon et al. [24] have identified global hypermethylation, it is most important to understand precisely where the methylation occurs, and what functional outcome has either already been studied or might be predicted.

Three of the most well-studied environmental factors of risk, and selected genes of risk with which they interact, will be the focus of this review: (1) The potential interaction between latitude, season-of-birth and the melanin-concentrating hormone receptor (MCHR1) (2) The use of cannabis and its potentially indirect interaction with AKT serine/threonine kinase 1 (AKT1), a phosphokinase that is part of the dopamine signaling cascade and (3) The interaction between environmental stress-induced increases in cortisol and an enzyme which initiates the kynurenine pathway, tryptophan 2,3-dioxygenase (TDO2). Though all three of the genetic loci that are featured have contributed to a significant polygenic risk score matrix for schizophrenia [21], these examples are intended to provide models of the epigenetics of psychosis, rather than to be the final word on the specific gene-environment interactions capable of causing psychosis. The literature on how the environmental factor is associated with psychosis risk and shows evidence of interacting with the gene in question will be presented first, followed by genetic association studies on the risk of psychosis conveyed by the gene, and finally, sites of epigenetic modification sites in the gene that may be of relevance to psychosis, transcription factors that interact with those sites, and confounders that must be considered when evaluating the study outcome.

An alphabetical list of acronyms used in this review, along with their description and any affiliated websites, can be found following the Discussion section. Unless otherwise specified, the many SNPs known to exist at the CpG sites reviewed in this manuscript are too rare to be of relevance to psychosis. 

## 2. MCHR1

### 2.1. Background for Environmental Impacts on the MCH System and Findings Relevant to Psychosis

The melanotropins are a family of peptides profoundly responsive to cues from the physical environment, originally investigated in fish for their pronounced effects on pigment dispersion in response to changes in background light levels [25,26,27], with the peptide alpha-melanocyte-stimulating hormone (alpha-MSH) increasing pigment dispersion and melanin-concentrating hormone peptide (MCH) diminishing dispersion. Mammalian MCH was cloned by Nahon et al. [28] and found to be abundantly expressed in the dorso-lateral hypothalamus. 

Whereas the alpha-MSH melanotropin system is closely tied to seasonal photoperiod responses in Siberian hamsters and other rodents [29], no response to photoperiod can be detected in the MCH system in those species, and there is only one report of a possible photoperiod effect of long days to depress MCH expression in higher animals, i.e., in sheep [30]. Rather, the primary seasonal cue for activating the MCH system may be temperature, as even short-term exposure to cold can increase the expression of the mRNA for MCH peptide [31]. Increased MCH function in response to colder environmental temperatures could be tied to conservation of energy stores in winter months, based on the involvement of mammalian MCH in creating and preserving fat stores [32,33], and the strong seasonal trend of this effect [34]. Thermogenesis from fat occurs primarily from brown adipose tissue (BAT), including in humans [35]; expression of MCHR1 has been demonstrated in BAT [36]; and Izawa et al. [37] have shown that activation of MCH producing neurons negatively regulates energy expenditure in murine BAT tissue. Consistent with this theme, prior work demonstrated chronic infusion of MCH in wildtype mice causes not only a decrease in energy expenditure, but also a lower body temperature [38]. These findings could be of particular relevance to the altered thermoregulation by individuals with schizophrenia, and the fact that they frequently overdress, even in summer [39,40]. A study of patients both on and off neuroleptics illustrated significantly lower body temperature in cases as compared to controls [41], and basal metabolic rate is apparently altered as well [42]. This characteristic of the disease appears to have long predated the advent of neuroleptic use [43]. 

Temperature plays a key role in the regulation of slow-wave and REM sleep by MCH neurons and although these neurons also release other sleep-related neuropeptides and neurotransmitters, the triggering of REM sleep by warmer temperatures (but within the thermo-neutral range) is likely dependent on MCH, because temperature-related prolongation of REM cannot be observed in MCHR1 knockout mice [44]. The important role of MCH in sleep is noteworthy given that a key feature of schizophrenia is the dysregulation of sleep, with a fairly consistent correlation between increased symptom severity and decreased latency to REM [45].

The overarching role of the MCH system has been described by Diniz and Bittencourt [46] as “integrative…converging sensory information and contributing to a general arousal of the organism”. As first discovered by Berbari et al. [47], MCHR1 is primarily located in neuronal cilial structures in the brain. Diniz and his colleagues view the predominantly cilial expression of MCHR1 in many brain regions as being consistent with a role in volume transmission, i.e., the sensing of extracellular neurochemicals not confined to the synaptic space. They have mapped MCHR1 expression across the mouse brain, finding that the receptor protein is expressed in cilia located in key subcortical, limbic and cortical regions, frequently co-localized with expression of the dopamine synthesis marker, tyrosine hydroxylase [48]. Although this co-localization did not occur in the substantia nigra, it was present in the parabrachial pigmented nucleus of the ventral tegmental area, a region that projects to the dorsolateral region of the prefrontal cortex and is considered to be a component of the dopaminergic circuitry perturbed in schizophrenia [49]. 

An initial investigation of how the melanotropin system might be relevant to the psychosis of schizophrenia was conducted in the early 1990s by Miller et al. [50], acting on the premise that hormonal responses to the seasons might underlie the epidemiological findings that the season of birth affects the risk of developing the disorder [51,52,53] and rates of schizophrenia appear to be elevated in higher latitudes, demonstrated in an analysis of independent studies [54]. Thus, an investigation of intra-cerebroventricular administration of MCH and alpha-MSH was undertaken in an animal model of the disrupted auditory sensory gating phenotype observed in individuals with schizophrenia. The results demonstrated the functional antagonism between the peptides first observed in teleost fish [25,26,27] extends to auditory gating, with MCH disrupting auditory gating in dose-responsive manner and eliciting a pattern observed in schizophrenia subjects, whereas alpha-MSH enhanced auditory gating beyond what was typically observed in controls [50].

Consistent with a change in sensory gating relevant to psychosis, some researchers have identified related traits affected by MCH in such a manner that would be expected to impair sensory gating, particularly anxiety and panic [55]. Infusion of MCH was shown to induce anxiety-like responses in an animal model whereas MCH antagonists have been found to be anxiolytic [56,57,58] and MCHR1 knockout mice are reported to exhibit lower levels of anxiety-like behavior [58,59]. Yet, in contrast to what would be expected based on these findings for auditory gating and anxiety, recent work in another model of sensory gating, pre-pulse inhibition (PPI) found that MCH knockout mice exhibited impaired PPI, though only in male mice at the highest decibel tested [60]. Such conflicting outcomes between auditory gating and PPI studies have been observed by others [61], illustrating that these tests likely measure a different component of sensory processing.

### 2.2. Genetic Linkage and Differential Methylation of MCHR1 Identified for Schizophrenia and Bipolar Disorder with Psychosis

The first genetic evidence that an association might exist between the MCH system and schizophrenia was published in 2006 by Severinsen et al. [62], showing an association between MCHR1 (previously known as GPR24) and schizophrenia plus bipolar disorder in the isolated Faroe Island population, with a less robust finding for alleles in their study subjects from mainland Scotland. The direction of risk was different between the two populations, pointing towards risk from opposite haplotypes (Figure 1). For the Faroe Islander isolate, having descended from predominantly Norse migrants over a millennium ago [63], a likely founder effect in allele frequencies of controls complicates a direct comparison with the mainland UK. However, within the island population, the comparison of cases and controls shows a striking 1.6-fold higher rate of the T allele (a synonymous mutation) in the CpG site rs133073 in cases (Figure 1 and Figure 2A). The T allele is closely linked to a G allele in rs133072 in the promoter region (Figure 1 and Figure 2A), which creates a CpG methylation site within an expression-activating SP1 transcription factor motif, but would not be expected to affect SP1 binding [64]. Nevertheless, adjacent SP1 motifs might be affected by methylation at this CpG site [64]. To summarize, the relative lack of methylation control of exon 1 in the Faroe Island cases could be viewed as part of an evolutionary tradeoff for a lack of methylation control of the promoter region in the Scottish cases, both being consequential for schizophrenia, with the risk either obscured or accentuated by gene–gene interaction in locus-specific manner [65]. Of the two outcomes, the relative lack of methylation control of the first exon would be predicted to have a more pronounced effect on the resulting gene expression [16]. Furthermore, Stepanow et al. [66] have shown that the degree of methylation is higher for the AC haplotype of rs133072-rs133073 than for the G-T haplotype, with the difference being greatest for those in the age range of greatest risk for schizophrenia onset, 20 to 30 years of age.

A follow-up genetic association study for schizophrenia and bipolar disorder in a predominantly Caucasian US cohort was carried out on one of the CpG loci identified in the risk haplotype, rs133073, along with single nucleotide polymorphisms (SNPs) in 4 other candidate genes [65]. Although the association with rs133073 did not reach significance as a single locus, the allele (T) identified by Severinsen et al. [62] as conveying risk for the Faroe Island population was found to contribute risk as part of a complex genotype with two of other candidate genes, MCHR2 and HM74, i.e., HCAR3, a gene also involved in regulation of fat stores. Demontis et al. [67] subsequently published a follow-up study on a Danish cohort of patients with schizophrenia, determining that out of three of the MCHR1 SNPs previously studied, all reached significance in their association analysis in a manner consistent with the results from the Scottish population previously studied, but only the rs133073 C allele held significance after correction for multiple comparisons. 

Two more distal SNPs have been found to be significantly associated with bipolar disorder and schizophrenia [68,69]. The most likely haplotypes with respect to rs133073 for all the other MCHR1 alleles of risk identified are shown in Figure 1. It is notable that of the nine SNPS in the risk haplotypes from the association studies to date, four are potential methylation sites.

**Figure 1 biomedicines-10-00561-f001:**
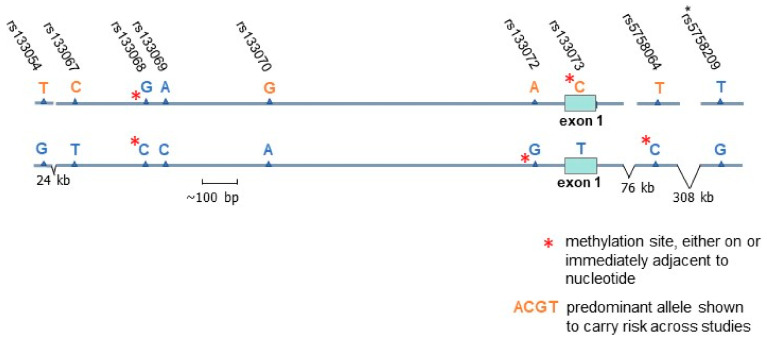
SNPs in linkage disequilibrium with the human MCHR1 gene that have been shown to exhibit various degrees of association with psychotic disorders, arranged as haplotypes with respect to rs133073 as determined by in silico analysis of the CEU population database (LDpop). Note that the allele of risk for *rs5758209 was not specified in the study by the Schizophrenia Working Group [68]. The alleles of risk from two small studies and one large study—in orange font—cluster on the upper haplotype and correspond to combined results from: a Scottish population, Severinsen et al. [62]: rs133068-70, -73; a Danish population, Demontis et al. [67]: rs133054-70, -72-73; a population of predominantly mixed European ancestry, Mullins et al. [69]: rs5758064; and a population of predominantly mixed European ancestry, Schizophrenia Working Group [68]: *rs5758209. Where the risk data from the Scottish population and the Danish population did not match, both alleles are coded blue. The risk alleles identified in a Severinsen et al. study [62] for the far northern population isolate on the Faroe Islands, are shown on the lower in silico haplotype for rs133068-70,-73. A founder effect may have resulted in a higher prevalence of the C allele for rs133073 in the controls studied in this isolated population, 50% as compared to the rate the authors reported for Scotland (41%) and in current LDpop data for the UK (36%), but it is likely that the haplotype containing the T allele was selected for over time, with the evolutionary tradeoff being more schizophrenia cases, becoming markedly more prevalent (1.6-fold) in the Faroe Island cases than in controls.

These findings relevant to CpG sites set the stage for research that might identify differential methylation in MCHR1 in cases with psychosis versus controls. The first such study did not seek to investigate MCHR1 specifically, but discovered its differential methylation as part of a genome-wide study in blood samples from 22 identical twin pairs in the UK [70], all Caucasian except for one twin pair, and discordant for schizophrenia or bipolar disorder, some with psychosis. MCHR1 (originally designated GPR-24) was the most significantly differentially methylated gene in the bipolar disorder (BPD) cases, 90% of whom carried a type 1 diagnosis, often associated with psychosis. The BPD cases exhibited significant hypomethylation at a CpG site (cg21342728), only 22 bp in the 5-prime direction from rs133073. Both cg21342728 and rs133073 reside in the first exon of the two exons in the gene (Figure 2). A later study carried out by Liu et al. [71] found that the same site, cg21342728, was significantly hypomethylated in psychosis cases diagnosed as schizophrenia, this time in blood samples from mixed race subjects in the US, 94 with schizophrenia as compared to 106 controls.

Demethylation of cg21342728 has been shown to lead to increased expression of MCHR1 [72] and in general, hypomethylation of the first exon would be expected to result in increased expression of most genes [16,17]. Yet, in a large, collaborative study utilizing deep sequencing technology applied to mRNA samples isolated from brain tissue of schizophrenia cases versus controls [21,60], MCHR1 expression was significantly decreased in the cases. Although the Vawter et al. study [60] adjusted for age, sex, postmortem interval, RNA integrity number (RIN), and brain bank, and Liu et al. [71] adjusted for age, sex, race, and alcohol use, while Dempster et al. [70], studying twins, corrected for multiple comparisons only, none of the analyses corrected for smoking. Individuals with psychosis are much more likely to smoke cigarettes and to smoke more heavily, than controls [73,74], and in a genome-wide study of differential methylation in smokers, MCHR1 was found to be significantly hypermethylated at two loci [75], one notably in the promoter region and one in intron 1 (Figure 2). Thus, it is of concern that the decreased expression in schizophrenia cases [60] may have been confounded by smoking-induced hypermethylation of a site in the promoter region of MCHR1 (Figure 2A). Although adjustment of the data set utilized a “surrogate variable correction” [21], this indirect statistical correction can never match direct findings for smoking and specific loci.

Methylation can also control alternative splicing [76,77], of particular relevance to the second MCHR1 isoform listed in Ensemble, spliced as shown in Figure 2B, but also of potential relevance to the availability of the putative alternative translation start site depicted in Figure 2A just downstream of a 5′UTR CpG. The 3′ ends of alternative splice sites are enriched in methylation (converted to 5hmC) in genes with highly variable methylation in the brain [78]. This alternative splicing scenario is almost certainly the basis for the alternative splice site in exon 2 (Figure 2B). A similar splice in the vicinity of the 5′UTR CpG would likely eliminate the alternative ATG site, and potentially the canonical ATG site as well (Figure 2), as observed in the NCBI entry for the MCHR1 transcript AY745811.1. However, the alternative transcripts may not be significantly differentially expressed, as only the total expression of MCHR1 is significantly different in cases versus controls and no single isoform of MCHR1 reached significance [21]. 

Immediately in the 5′ direction of both these CpG sites is a motif for HNF-1 (predicted by the well-cited program Alibaba2), a transcription factor known to regulate tyrosine hydroxylase [79], the enzyme often co-localized with MCHR1 in immunohistochemistry of the mouse brain [48] and a marker for dopamine synthesis. This finding raises the strong possibility of co-regulation of MCHR1 and tyrosine hydroxylase. There is only one other motif for HNF-1 in MCHR1, located in intron 1, but not adjacent to a CpG site.

**Figure 2 biomedicines-10-00561-f002:**
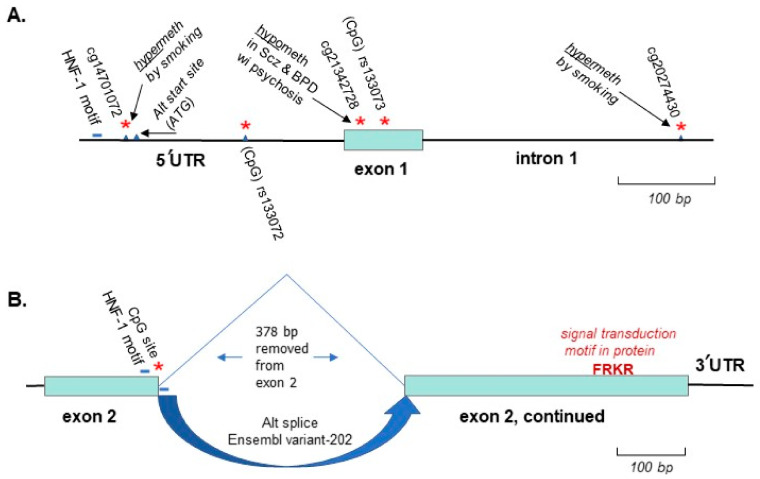
Methylation sites in human MCHR1 of likely functional significance (marked by red asterisks). Just upstream of two CpG sites are binding motifs for the transcription factor HNF-1, and though not abundantly expressed in the brain [80], is prominently involved in regulation of tyrosine hydroxylase [79], an enzyme often co-localized with MCHR1 in the brain [48]. (**A**). Shown are the 5′UTR, exon 1, and a portion of intron 1 of the primary isoform, predicted in Ensembl to yield a 353 amino acid (aa) protein product, along with two CpG sites associated with SNPs (in the promoter region and in exon 1), and two CpG sites reported to be hypermethylated by smoking [75]. The cg21342728 site in exon 1(located by iMethyl) is hypomethylated in psychosis cases [70,71], and the SNPs that exist at this site are far too rare to be relevant for psychosis. Just downstream (21 bp) from the 5′UTR CpG site lies an alternative ATG start site predicted from the nucleic acid sequence entered for the 5′UTR, first exon and first intron (NCBI ORF finder) yielding an ORF in-frame with Exon 1, and 422aa in total protein size, more closely matching the size observed in mouse brain as detected by an antibody directed towards a 16aa epitope in exon 1of the primary isoform (https://www.alomone.com/p/anti-melanin-concentrating-hormone-receptor-1-extracellular/AMR-041, accessed on 1 November 2021), although Saito et al. [81] have shown glycosylation contributes substantially to the higher MW bands. Alternative splicing near the 5′UTR CpG site, as shown for CpG sites in other genes [76,77], could potentially explain control of the use of the alternative upstream translation start site and the canonical downstream site, as documented in one MCHR1 transcript submitted to NCBI, AY745811.1. (**B**). The second protein-coding MCHR1 isoform listed by Ensembl involves splicing out an in-frame sequence in exon 2 immediately following the G nucleotide in the CpG site shown. The length of this isoform would be 227aa with retention of the FRKR aa motif thought to be a component of signal transduction in this receptor [82], the only sequence difference versus the primary isoform being the missing residues. For the purposes of locating this CpG site, the relevant SNPs are rs369827677 and rs752123648.

Table 1 summarizes the key points on MCHR1 relevant to the epigenetics of psychosis. Particularly noteworthy is the fact that many of the variable genetic linkages identified have to do with methylation sites. This topic will be examined further in the Discussion section.

## 3. AKT1

### 3.1. Background on Cannabis-Induced Psychosis and the AKT1 Gene as a Model System for This Environmental Effect

Cannabis is regarded as among the top five environmental agents that have consistently been associated with increased risk for long-term psychotic outcomes and tops the list in terms of effect size [83], increasing risk up to 5-fold for daily users of moderately high strength product in epidemiological studies [84,85]. No other environmental factor or gene of risk comes close to imparting this effect size when assessed as a single agent. Administration of moderate doses of the psychoactive ingredient of cannabis (THC) in the clinic elicits psychotic symptoms in individuals without a family history of psychosis [86,87,88]. In Denmark, a country with rigorous tracking of health metrics in its population, the increase in cannabis use and cannabis use disorder has been mirrored by an overall increase in schizophrenia cases [89]. 

Exploration of the role of the active ingredient of cannabis, Δ^9^- tetrahydrocannabinol (THC), in models of traits of psychosis has been informative. In studies investigating the sensory gating traits seen in psychosis, the administration of THC to rats disrupts auditory gating in a manner consistent with what is seen in individuals with schizophrenia [90], confirming prior work that administering THC or other cannabinoid receptor agonists to rodents disrupts auditory gating [91,92]. Of note, the auditory gating paradigm in general shows good reproducibility based on a recent meta-analysis [93]. 

One candidate mechanism for the impact of cannabis on psychosis is its effect to increase dopamine release in key brain regions [94]. The dopaminergic system has long been considered to play a strong contributory role in psychosis, primarily because most effective antipsychotics act as antagonists at dopamine receptors [95], but also because a few different agents which increase dopamine concentrations at the synapse can trigger psychosis [96,97,98,99]. The gene AKT1 has attracted much interest in this regard based on its role in the dopamine signaling cascade (Figure 3). Binding of dopamine to the dopamine receptor DRD2 inactivates AKT1 through dephosphorylation [100,101,102], which in turn leads to activation of a signaling molecule under inhibitory control by AKT1, GSK3β. As a phosphokinase, AKT1 is networked with numerous other signaling cascades, performing a wide range of functions in the brain, including regulating the size of neurons and their survival [101]. 

The expression studies utilizing postmortem brain tissue as well as blood samples from patients with schizophrenia have been mixed, identifying downregulation of AKT1 protein and mRNA expression in psychosis [103,104], but higher mRNA expression in treatment-resistant-schizophrenia [105] and in methamphetamine-induced psychosis [106] as compared to controls. Furthermore, Karege et al. [107] did not find that the protein expression level differed between controls and schizophrenia cases as a broad group, and Chaumette et al. [108] also reported that the mRNA expression was not significantly different in a longitudinal study of conversion to psychosis. Nevertheless, Gandal et al. [21] identified two alternative transcripts significantly upregulated in schizophrenia, a short transcript of 663 bp (labeled AKT1-15) that was significant below the false-discovery rate (FDR) threshold and the 26,402 bp primary transcript (labeled AKT1-001) that trended for significance.

### 3.2. Genetic Linkage of AKT1 to Psychotic Disorders, Gene-Environment Interaction with Cannabis Use and Potentially Relevant Methylation of AKT1 Identified

The initial report of a genetic association between SNPs in AKT1 and schizophrenia was published by Emanian [103], and evaluated in subsequent studies, but with inconsistent results. The odds of the A allele of risk in rs1130233 being associated with psychosis was reported to be 2.5-fold in some schizophrenia populations [109], yet no linkage was found by others [110]. 

Acting on the established effect of cannabis to increase dopamine release and the role of AKT1 in dopamine signaling, Di Forti and colleagues pursued a study investigating a gene-environment interaction between cannabis use and the SNP rs2494732 of AKT1. Neither this SNP nor rs1130233 are CpG sites. Although they failed to replicate a significant association with psychosis in the absence of cannabis use, they did find a significant interaction with cannabis to increase psychosis risk in homozygous carriers of the C allele [111]. Subsequently, Bhattacharyya et al. [112] reported that the A risk allele in rs1130233 identified by Tan et al. [109], conferred sensitivity to cannabis induced deficits in psychomotor control. Remarkably, rs1130233 and rs2494732 are in linkage equilibrium in populations of African descent (i.e., functionally unlinked for genetic association purposes according to LDpop) but are significantly linked in European population, either pointing towards an evolutionary bottleneck during migration [113] or intense evolutionary pressure [114] following migration.

Further study of cannabis interaction with rs2494732 in AKT1 in combination with a dopamine receptor (DRD2) allele was carried out by Colizzi et al. [115], but unfortunately, the proportions of individuals with African descent were higher than individuals of European descent in their case population and the reverse was true for the control group. In this regard, the difference in linkage equilibrium between rs2494732 and rs1130233 in these populations becomes important, Thus, based on the significant linkage between the two SNPs in the European populations but not in those of African descent, it turns out that the expected prevalence of the protective alleles being on the same haplotype would be somewhat higher in the study controls while the expected prevalence of the *risk* alleles being on the same haplotype would be *much* higher in the controls. Thus, adjusting for the effect of race on rs249732 allele frequencies does not correct for this haplotype-specific effect. For this reason, as well as population stratification issues that are magnified by racial differences between groups, it is unwise to compare cases and controls with such a large racial imbalance [116].

A recent investigation of the effect of methylation at AKT1 CpG sites on behaviors elicited by THC administration [117], determined that the number of A alleles in rs1130233 (i.e., 2, 1 or 0) positively correlated with preexisting, downstream CpG methylation levels (Figure 4), a plausible connection because alleles at key sites have been shown to differentially recruit methylation activity for other sites within a gene [118]. A motif for the transcription enhancing factor SP1 (predicted by the program Alibaba2) lies in the region of the differential methylation (Figure 4), and although ^m^CpG methylation within SP1 motifs does not affect SP1 binding, methylation outside of the SPI binding motif [64] or double methylation leading to ^m^C^m^CpG on the antisense strand, does inhibit binding [119,120]. Such a CCG sequence is located on the antisense strand in the middle of the SP1 binding motif in exon 10 (Figure 4). When the authors compared the overall methylation level in this region with behavioral brain activation patterns in brain scans, they found increasing methylation levels correlated with activation patterns during fear processing and levels of anxiety in healthy controls. Among other brain regions activated in tandem with differential methylation were the superior frontal gyrus and the anterior cingulate cortex. Of potential relevance, a projection between these two brain regions has been found to be underdeveloped in longitudinal studies of adolescents who are regular cannabis users [121]. 

Once again, however, an apparent failure to address smoking behaviors complicates the interpretation of epigenetic results. Research has very consistently identified global hypomethylation of AKT1 in smokers [75,122,123,124,125]. This raises the possibility that smoking versus nonsmoking was controlling the degree of methylation observed to be correlated with fear processing and anxiety. Many of the subjects in the Blest-Hopley et al. study of AKT1 [117] were smokers, and although they had been advised to refrain from nicotine for 4 h prior to the study onset, that is unlikely to be enough time to reverse hypomethylation patterns induced by their tobacco habit [122]. Behavioral tests of nicotine’s effect on fear in animal models show an effect that is highly dependent on whether the animal has been exposed to acute or chronic nicotine treatment and if chronic, whether they have entered withdrawal or not. Mice subjected to chronic treatment with nicotine can show reduced responses to the fearful stimulus if tested several hours later [126] and human subjects who are chronic nicotine users have similarly shown a reduced ability to discriminate between fearful and neutral cues [127]. In summary, it appears likely that the subjects with a lowest degree of methylation near rs1130233 may have been those that smoked the most heavily, and because of their nicotine habit, were also the most likely to exhibit reduced activation of the fear response. Such an outcome would indicate the effect of THC on fear processing was blunted by smoking tobacco, an additional finding and important in its own right.

**Figure 4 biomedicines-10-00561-f004:**
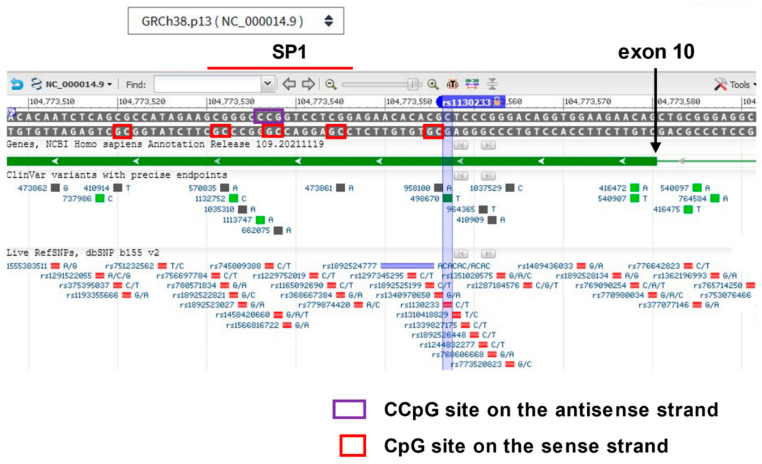
Screenshot of exon 10 of the AKT1 gene on the NCBI gene map for chromosome 14, intended to illustrate the power of NCBI data available, including the panoply of SNPs shown below, most very rare. Note the reverse orientation of the gene. Hovering a cursor over exon 10 (shown in dark green) will allow access to the FASTA nucleotide and protein sequence for the entire gene. Location of rs1130233 is shown relative to downstream CpG sites identified by Blest- Blest-Hopley et al. [117] to be differentially methylated in correlation with degree of THC-induced regional human brain activation in response to a fearful cue. The chromosomal location given in their paper for two sites found to be most significantly differentially methylated, chr14: 104,773,527–104,773,522, does not correspond to any two adjacent CpG sites separated by 5 bp. However, just upstream are two methylation sites separated by 5 bp, are part of a binding site for the expression-enhancing transcription factor SP1, according to the in silico analysis tool Alibaba2. Binding of SP1 within an exon is known to occur for other genes [128,129,130]. SP1 binding is generally not affected by methylation, except when the sites are immediately outside the binding motif [64] or include double methylation to yield ^m^C^m^CpG inside the binding motif on the antisense strand [119,120]. In these scenarios, SP1 binding is inhibited by methylation, and expression of the transcript may be similarly affected. Blest-Hopley et al. [117] did not make clear if the cg probes they utilized would pick up ^m^C^m^CpG. Regardless, the global hypomethylation of AKT1 associated with smoking may well act to restore SP1 binding in exon 10.

No differential methylation of AKT1 relevant to psychosis risk has been identified in other studies reported on MethBank. What has not been observed in the publications cited here is the striking structure of the AKT1 in the 5′UTR region, along with three additional CpG sites in the first exon (coverage of 222 bp in total). The CpG island predictor EMBOSS rates the 5′UTR region as exhibiting “islands of unusual CpG composition” with a CG content much greater than expected. Unfortunately, none of these CpG sites are common polymorphisms, so there is no variable linkage to alleles in rs1130233 or rs2494732 that would be informative. Contrary to expectation, CpG islands are most often unmethylated, and commonly found in the promoter region of housekeeping genes [131].

A summary of this review of AKT1 can be found below (Table 2).

## 4. TDO2

### 4.1. Background on Relevance of the Kynurenine Pathway to Psychosis and the Response of the TDO2 Gene to Relevant Environmental Stimuli

The kynurenine pathway (Figure 5) primarily evolved to fulfill two major roles dictated by environmental conditions, the regeneration of nicotinamide adenine dinucleotide (NAD+ and NADH) from tryptophan under conditions of a vitamin B3 deficiency [132,133] and as an essential component of the innate immune system, regulating everything from immune rejection processes that must be controlled for successful pregnancies, e.g., González et al. [134], to being part of the first line of defense against pathogens before an antibody response can be mounted [135]. Against some organisms, such as *Mycobacteria tuberculosis*, acquired immunity is not as important as innate immunity in preventing fulminant infection of the lungs [136] and the innate immune response is what must prevail.

Two enzymes catalyze the initiating step of the pathway, the tetramer tryptophan 2,3-dioxygenase (TDO2) and the monomer indoleamine 2,3-dioxygenase (IDO). Although their roles can overlap to a certain degree, TDO2 is likely more important for regenerating NAD+ (Figure 5), being negatively regulated by the availability of the end product, nicotinamide, and possessing a higher activity rate constant than IDO [137], important because without NAD+, physiological functions cease. IDO has been regarded as more important for responding to infection, being strongly activated by interferon ɣ (IFNG), as well as other cytokines [138].

An indication that the kynurenine pathway might have something to do with psychosis was the longstanding observation that pellagra, a dietary disorder brought on by an NAD+ deficiency activating kynurenine synthesis, can cause psychotic delusions that are reversed by supplementation with the vitamin B3s [139,140]. One of these B3 vitamins, nicotinamide, exerts negative feedback control on TDO2, and NADH or NADPH are even more potent regulators [132,133]. In addition, there are a host of other conditions which activate the pathway and trigger psychosis. Infections by a variety of microorganisms associated with psychotic episodes have been shown to be associated with upregulation of the pathway [141,142,143,144,145,146,147,148,149,150,151], primarily via cytokine stimulation of IDO gene expression and activity. Because kynurenine crosses the blood–brain barrier [152], even kynurenine synthesized in the periphery during infections can affect brain function. 

An intermittent heme synthesis disorder known as porphyria is associated with psychosis [153,154]. Both TDO2 and IDO require heme for catalytic activity and cyclical restoration of TDO2 and IDO catalysis by heme availability would be expected to lead to markedly increased flux through the kynurenine pathway stemming from upregulation of the enzyme monomer synthesis as a compensatory mechanism during the heme deficiency phase, as observed by Wetterberg et al. [155] in an animal model. 

Corticosteroids upregulate the pathway via stimulation of TDO2 transcription [156], and this will form a major emphasis for the analysis of epigenetic effects on TDO2 in the next subsection. Steroid-induced psychosis is well documented [157], and its association with kynurenine levels is certainly worthy of more study.

The first direct demonstrations of kynurenine upregulation in psychotic disorders were published in 2001, finding that kynurenine and its metabolite, kynurenic acid, were increased in concentration in postmortem brain tissue from individuals with schizophrenia and in CSF of living patients [158,159], subsequently confirmed by other studies [160,161,162]. Miller et al. [160,163] identified TDO2, rather than IDO, as the likely source for the increase in metabolites, as its expression was elevated in postmortem cortex of individuals with schizophrenia at both the mRNA and immunohistochemical level, specifically in the astroglial cells of white matter. The PCR products were cloned and sequenced, revealing >99.9% homology with the published sequence for TDO2. In several of these studies, the potentially confounding impact of smoking was examined and determined to be unassociated with the increases observed. The potentially confounding role of antipsychotic medications was also tested by comparing patients who were off their medications near the time of death to those on medication at death [160,161,163], and the trends did not support an effect of antipsychotic medications to stimulate the kynurenine pathway, similar to what was observed in an animal model of antipsychotic drug administration [158].

The increased expression of TDO2 mRNA and protein in the frontal cortex (Brodmann’s area 10) and in the anterior cingulate cortex of individuals with psychosis [160,163] was complicated by the finding that Western blots of normal control tissue from two unrelated individuals, using the antibodies designed to respond to two different epitopes (termed “A” and “B”), each uniquely recognized different isoforms of shorter length (~32 kD and ~35 kD, respectively; Figure 6) than the expected 47 kD, full-length TDO2. Schmidt et al. [164] studied the specificity of antibody TDO2A and confirmed that it recognized full length human recombinant TDO2 expressed in T-REx™ HeLa cells, and Opitz et al. [165] determined the antibody recognized the full length protein in glioma tissue. The likely origin of TDO2B was experimentally determined as shown in Figure 6A. Epigenetic control of the two isoforms, TDO2-A and TDO2-B, is probable, but has not yet been studied.

A more recent investigation has reported increased expression of TDO2 in cases of psychosis [21], though just trending for significance above the FDR, and the increase in TDO2 expression has also been confirmed for both high- and low-level cytokine groups of psychosis patients [166]. Studies of the related conditions of bipolar disorder and of suicidal behaviors, which too often develop in cases of schizophrenia [167,168], have identified kynurenine pathway activation in spinal fluid [169,170] or in blood samples, more consistently so when appropriate methodology is used in blood samples [171,172], i.e., avoiding the use of trichloroacetic acid (TCA) to precipitate protein prior to analysis for tryptophan and kynurenine metabolites. The use of TCA is problematic for blood because the variable hemolysis can result in variable amounts of iron present, a catalyst for the non-enzymatic formation of kynurenine from tryptophan under acidic conditions [160].

In addition to the other environmental stimuli mentioned above, the season of birth finding in schizophrenia may bear some relationship to seasonal regulation of the kynurenine pathway. As discussed in the previous section on melanotropins, the literature supports a significantly greater risk of developing schizophrenia if born in the late winter or early spring, along with a fall deficit in such births [51,52,173]. In animal studies, it is seasonal photoperiod effects in the third trimester that have been shown to have the most impact [174,175] and which can lead to enduring impacts on behavior and development [176,177]. Remarkably, a recent study of pregnant women carried out by Levitan et al. [178] detected seasonal variations in their blood kynurenine levels, such that sampling conducted in winter of mothers who gave birth in the spring revealed an elevation in kynurenine as compared to their early pregnancy values, and sampling conducted in summer of mothers who gave birth in the fall, detected a decrease in kynurenine as compared to their early pregnancy values, particularly pronounced for those mothers with Seasonal Affective Disorders (SADs). In contrast, tryptophan levels fell between early and late pregnancy sampling for both the spring birth and fall birth cohorts. Unfortunately, the method employed for precipitating protein from the plasma prior to analyzing for kynurenine and tryptophan was not specified in the paper.

There is very likely a role for sex hormones in the environmental modulation of psychosis, as males are at greater risk for this disorder [179], and the risk for psychotic disorders increases in the peri- and post-menopausal time [180]. Of the three genes in this review, it may be TDO2 regulation that illustrates some degree of relevant gender-specificity. Badawy [181] reports that estrogen directly inhibits the activity of the enzyme by affecting its ability to bind heme, although this is a finding not yet confirmed by other groups. In addition, studies show upregulation of TDO2 expression seen in estrogen-receptor-negative but not in estrogen-receptor-positive breast cancer [182], suggesting that expression of TDO2 may be under negative regulatory control by estrogen in this setting. 

Stress, particularly childhood adversity [183], has been found to increase the risk of psychosis, and acute stress in late teens or early adulthood is associated with psychotic breaks [184]. Hubbard and Miller [185] conducted a meta-analysis revealing that cortisol levels are generally elevated in first episode psychosis. Cortisol acts to upregulate TDO2 expression [156], and an epigenetic effect in TDO2 associated with the stress of childhood adversity has also been identified, as covered in detail in the next section.

**Figure 6 biomedicines-10-00561-f006:**
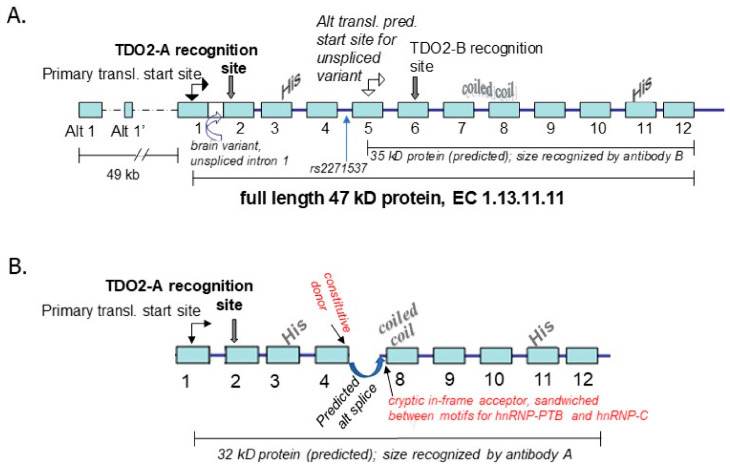
Isoforms of TDO2 in the brain, recognized by antibodies directed to different epitopes, TDO2-A and TDO2-B; the specificity of antibody A for TDO2 confirmed by Schmidt et al. [164] in a Western blot of recombinant human TDO2. Epigenetic regulation of these isoforms is likely. (**A**). Experimental evidence derived from sequencing a cDNA clone of a brain transcript that retained intron 1, supports the existence of isoform TDO2-B, as two ORF finding programs (™DNAstar, NCBI) revealed this intron-1-retained sequence shifts the predicted start site to an ATG in exon 5; the resulting protein would be 35 kD as seen in the Western blots. The location of the SNP for a genetic association study, rs2271537 [65] is indicated. (**B**). In silico prediction of one potential identity for TDO2-A. As the post-translational modifications of TDO2 have to-date been shown to be limited to phosphorylation of 6 residues (see dbPTM), the maximum likely kD difference between the predicted isoform from in silico analysis and the actual isoform in vivo is likely limited to <0.5 kD. Analysis of constitutive and cryptic donor/acceptor pairs by the program ASSP identified a single continuous ORF (confirmed by the program Augustus and the NCBI ORF identifier) that would match the 32 kD protein size recognized by TDO2-A, spliced as shown from a constitutive donor at the end of exon 4 to an in-frame cryptic acceptor—gactattaat—identified by the ASSP program in the 3-prime region of intron 7. Surrounding this 5′ acceptor site lie two motifs for alternative-splicing-repressors identified by the program Alibaba2, hnRNP-PTB (CTCTCT) and hnRNP-C (TTTTT), substantiating this site as an important regulatory point [186]. The missing sequence would modify the active site, as would the substitution of the in-frame aa sequence QTINAIFFLKA from intron 1. Note retention of the histidine (HIS) residue coded by exon 3 and a tryptophan residue coded by exon 11 which are thought to be important for holding the 5-membered ring of tryptophan in the active site, along with a phenylalanine coded by exon 3 (via interacting with the aromatic ring of tryptophan), whereas the heme is bound by histidine coded for in exon 11, a tyrosine coded in exon 6, and a glycine coded in exon 6 [187]. The coiled coil region coded in exon 8, thought to be important for tetramer formation [188], was predicted by both ™DNAstar and WAGGA).

### 4.2. Genetic Association Study Results for TDO2 in Schizophrenia and Bipolar Disorder with Psychosis, Key Regulatory Sites within the Gene, and Potentially Relevant Methylation Induced by Stress

An initial genetic association study of TDO2 in psychosis determined that although a SNP in intron 4 (rs2271537) was not significantly associated with schizophrenia and bipolar disorder as a single locus, a complex genotype illustrated significant gene–gene interaction of a homozygous form of this SNP (CC) with 2 other loci, the melanotropin receptor MC5R, and the melanotropin receptor MCHR2 [65]. This complex genotype increased risk for schizophrenia 4.3-fold, *p* = 0.016 and 3.3-fold, *p* = 0.046 for a combined group of schizophrenia plus bipolar patients, carried by 19% of patients studied, while the single loci alone carried no significant risk. 

In regard to epigenetic effects of relevance to stress and glucocorticoid regulation of TDO2, a prospective, genome-wide study of differential methylation detected a difference in TDO2 associated with indices of childhood adversity, including socioeconomic status [189]. A state of low socioeconomic resources during childhood has been shown to increase the risk of psychotic disorders [190,191,192], to such an extent that it is considered a potentially confounding variable and corrected for in studies of psychosis incidence [193]. The TDO2 epigenetic results associated with low socioeconomic level in childhood showed decreased methylation at a site in alternative exon 1 (cg15736994, Alt-exon 1; Figure 7), but to evaluate the net effect on gene expression when the Alt-1 exon is hypomethylated will require more research. Methylation within alternative exons can act to enhance inclusion in the final transcript [77], but the absolute level of expression may be lower when the methylation is in the first exon [16], thus the net effect cannot be predicted.

Of great relevance to the epigenetic interaction of stress with TDO2 is the presence of three glucocorticoid-response-elements (GREs) in the 5′ region for the primary transcript [194], which would serve to upregulate transcription of TDO2 when the glucocorticoid receptor/cortisol complex binds [195]. On the one hand, this direct impact of elevated cortisol from a stressful environment circumvents the need for regulatory control of TDO2 via methylation induced by stress. Yet, on the other hand, methylation within the sequence of GREs (Figure 7) has been shown to enhance binding of cortisol, a feature also found for a few other transcription factors [196]. However, no differential methylation of the GRE elements has yet been identified in cases of psychosis versus controls. Nearby are binding motifs for the transcription factor CEBP, known to foster glucocorticoid stimulation of transcription [197]. Kudo and coworkers [198] identified a particular CEBP-β motif lying within 116 bp of the translation start site, demonstrating it to be capable of constitutively upregulating TDO2 expression (Figure 7). 

Salivary levels of the glucocorticoid cortisol have been found to be significantly higher upon awakening in night-shift workers [199], and a genome-wide study of differential methylation related to disruption of circadian rhythms during pregnancy, nightshift work has been found to result in hypermethylation of a CpG site in intron 3 (cg03709468; see iMethyl for location) in placental tissue of women who had engaged in such work [200]. This methylation site is within an NF-kB motif (Figure 7), which has been shown to be involved in responding to stress and to loss of sleep [201]. Methylation of this NF-kB site would be expected to block NF-kB binding to the motif [202]. Binding of NF-kB to introns has been shown for other genes, e.g., ICAM-1, where its role is generally to stimulate expression [203].

As evaluated for the other genes of interest, the impact of smoking on TDO2 methylation and gene expression is important to determine. Ringh et al. [125] found that TDO2 mRNA expression was lower in smokers than in nonsmokers, consistent with what was observed for TDO2 mRNA expression by Miller et al. [160,163]. In a genome-wide methylation study of the impact of smoking during pregnancy on the methylation patterns seen in newborns, Sidkar and colleagues [204] found that a TDO2 site recognized by the probe cg04795662, 28 bp upstream of the ATG translation start site in exon 1 (see iMethyl for location) was hypermethylated in newborns of mothers who smoked. Such hypermethylation in the 5′ UTR outside of a GRE motif is most likely associated with a lower level of expression of the gene. 

**Figure 7 biomedicines-10-00561-f007:**
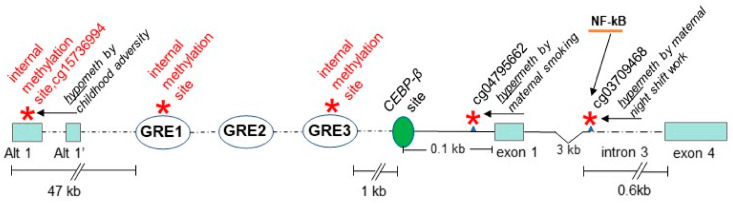
Location of key methylation sites in TDO2 (marked with red asterisks): within alternative exon 1 (Alt-1), in two of the three glucocorticoid motifs and just upstream of exon 1 in the promoter region. Methylation of the GRE sites should enhance binding of the glucocorticoid receptor protein, docking of cortisol to the site [195], and act to increase expression of the gene. The CEBP-β transcription factor motif identified by Kudo et al. [198] enhances GRE motif effects when both are bound by their respective ligands [197]. Hypomethylation within Alt-exon 1 (cg15736994; see iMethyl for location) is observed in children who are economically underprivileged [189], and would likely diminish inclusion of the alternative transcript from this location, as methylation within alternative exons can act to enhance inclusion in the final transcript [77]. However, the absolute level of expression may be higher when the hypomethylation is in the first exon [16], thus the net effect cannot be estimated. Hypermethylation of a CpG site just upstream of exon 1 was reported for newborns of mothers who smoked during pregnancy [204]. Chronic mild stress in adult rodents has been shown to decrease methylation in the general region of the promoter [205]. Hypermethylation at a CpG site in intron 3 is found in placental tissue of pregnant women who worked on the nightshift [200]. The transcription factor NF-kB is predicted by the program Alibaba2 to bind at this site, and among other roles, it is involved in response to immune stimuli as well as stress from sleep deprivation [201,206], certainly consistent with nightshift work. Methylation of this NF-kB site would be expected to block NF-kB binding to the motif [202]. Binding of NF-kB to introns has been shown for other genes, e.g., ICAM-1, where its role is generally to stimulate expression [203].

A summary of these results can be found below (Table 3).

## 5. Discussion

For each gene covered in this review, salient epigenetic features emerged that may have some bearing on psychosis, including CpG sites in close proximity to specific transcription factor or enhancer motifs. MCHR1 exhibited hypomethylation of exon 1 in psychosis subjects from two independent studies [70,71], while HNF-1 motifs were found adjacent to CpG sites thought to be associated with alternative splicing (Figure 2). HNF-1 is prominently involved in regulating the dopamine-synthesis marker tyrosine-hydroxylase [79], the expression of which co-localizes with MCHR1 in key brain regions [48], suggesting a degree of co-regulation of the two genes. For AKT1, a differentially methylated region associated with a psychosis-related behavioral response overlaps with a binding motif for the well-known activating transcription factor, SP1, and a specific pattern of methylation may interfere with its binding (Figure 4). For TDO2, CpG sites lie within two GRE motifs (Figure 7) involved in binding the receptor for the stress-related hormone, cortisol, known to be associated with onset of psychotic behaviors [184,185]. Genome-wide differential methylation studies identified other sites of interest in TDO2. Hypermethylation within an alternative exon-1 in the far upstream region found to be associated with childhood adversity [189] and within the gene body (Figure 7), a site hypermethylated in pregnant women engaged in nightshift work falls in the middle of an NF-kB binding motif [200]. NF-kB is a transcription factor demonstrated to be responsive to stress and loss of sleep in particular [206].

Whereas MCHR1 has repeatedly shown significant genetic linkage to psychotic disorders as a single gene, AKT1 and TDO2 have not in a reproducible manner, yet, that does not preclude these loci from exerting relevant effects as a result of environmentally-induced epigenetic changes. Some variability exists in the precise MCHR1 SNP most significantly associated with psychosis, as well as in which allele of the SNP the disease is associated with. A similar lack of coherence in the association of MCHR1 alleles with human obesity has been observed and attributed to differential methylation altering the function of the allele in question [66]. For example, when an individual is heterozygous at a CpG site, allele-specific expression can occur as the result of methylation [66]. Such allele-specific expression gives merit to the concern evidenced by Severinsen et al. [62] in constructing haplotypes of risk, and underscores the ways in which differential methylation caused by interaction with the environment can lead to a lack of reproducibility in genetic association studies. As this field becomes more refined, genetic association studies of psychosis may become targeted for specific environments in order to clarify the results.

A few themes were found in the literature which can assist in the general interpretation of epigenetic signatures, including: (1) methylation in the 5′UTR and in the first exon is generally associated with repression of expression [16,17], but with important caveats: (a) when the methylation occurs within a GRE element, it enhances binding of the glucocorticoid docking protein that binds cortisol [195], which in the case of TDO2, increases expression; (b) methylation within an alternative exon can act to affect its inclusion in the final transcript [77]; and (c) methylation can mark sites for alternative 3′ splice sites, repressing expression of the contiguous transcript but also creating a different product altogether [76,77,78]; and (2) methylation in the gene body—other than exon 1—is generally permissive for gene expression, particularly if 5hmC is generated from 5mC [19,207]. 

Evaluating the correspondence between gene expression and epigenetic results in cases and controls was limited by the available research, in particular when correspondence between changes in the brain and the blood has not been confirmed. The hypomethylation in exon 1 of MCHR1 for psychosis cases, documented by two independent research groups in blood samples [70,71], did not correspond well with the reduced brain expression of MCHR1 in psychosis cases [21,60], the latter outcome potentially attributable to smoking. The expression results for AKT1 were too variable to compare to any methylation data, and for TDO2, although the elevated differential expression results between cases and controls were very consistent [21,160,163,166], relevant methylation data for the pivotal regions of the promoter and the first exon were not available in the literature for psychosis cases versus controls.

When left uncorrected, smoking has emerged as a major confounding variable for both the epigenetic signature in psychosis and in the expression results. Yet, another way to view this variable is that the direction of effect may provide a clue as to the potential relevance of the gene to psychosis, should smoking prove to be a form of self-medication. If so, then the effect of smoking to decrease overall AKT1 methylation but to increase relevant TDO2 and MCHR1 methylation may indeed have functional relevance to psychosis. There is a great divide in the literature in regard to the effects of nicotine, with several well-conducted epidemiological studies pointing towards a causal role for nicotine in triggering psychosis, as reviewed by Quigley and MacCabe [208]. However, as the authors point out, a major confounder for the nicotine-psychosis association is that so many cannabis users also smoke cigarettes. In this regard, the prospective studies on which comes first, nicotine use or the psychosis, were not been convincing because they failed to correct for cannabis use, and unlike cannabis, the acute effect of nicotine administration in a clinical setting does not include psychotic symptoms. Dalack and Woodruff [209] found that withdrawal from tobacco smoking led to exacerbation of schizophrenia symptoms, and one epidemiological study reported a lower risk of schizophrenia in smokers [210], while clinical studies have found a benefit for nicotine in normalizing the auditory gating trait of psychosis [211,212]. 

In contrast to smoking, very little has been done in the way of genome-wide studies of the epigenetic effects of antipsychotic drugs. This is an oversight that will continue to hamper any conclusion about epigenetic changes in specific genes seen with psychosis. Many RNA expression studies have corrected for antipsychotic drug use where possible, in some cases utilizing brain bank tissue of patients who were never medicated as compared to controls and patients who were medicated at death. Of the genes covered in this review, there are publications on trends in TDO2 expression that point to a lack of confounding by antipsychotic drugs [160,161,163,166]. Although neither MCHR1, AKT1 nor TDO2 were part of the gene cluster shown to exhibit a major effect from dose-years of antipsychotic medication in the work by Gandal et al. [21] (see M21, Supplementary Table 5 of that publication), the information was not detailed enough to rely on for a specific gene of interest or for how the medications might affect isoform-specific expression. As for AKT1, a direct confounding effect should be assumed because most antipsychotic drugs block the DRD2 receptor, which would preferentially leave AKT1 in the active, phosphorylated state (Figure 3).

The highest degree of differential methylation found to be associated with psychosis, as identified in epigenome-wide-association studies (EWAS; [20,24]) does not necessarily reflect important biomarkers, as lesser degrees of differential methylation and/or expression in genes with more potent effects on psychosis can supersede greater methylation changes in genes of lesser effect. Additionally, as made all too clear for the genes reviewed here, where that methylation happens is paramount. Does it occur in transcription factor binding sites? If so, will the binding be enhanced or reduced? Is the methylation within the exon-1/enhancer/promoter region or is it in other components of the gene body, where methylation can enhance expression, often via conversion to 5hmC? Is there a double CCG methylation possible on the reverse strand that would affect SP1 binding in the gene body? Does the methylation occur at the 3′ end of a potential alternative splice site, perhaps fostered by an adjacent transcription factor motif as in MCHR1 (Figure 2)? If so, a different isoform may be the result. These site-specific effects can potentially explain why genes identified as being the most differentially methylated in psychosis [20] do not always match the genes that are the most differentially expressed in psychosis [21], and why the direction of the effect for methylation and the direction of effect for expression often reverse their association. Importantly, the Montano et al. study [20] adjusted for smoking. Another relevant distinction between these two studies was the former assessed methylation in blood samples and the latter in brain samples. Nevertheless, among the topmost differentially methylated in the Montano et al. study [20], there are some notable matches with significant differential expression in the Gandal et al. study [21]. The direction of the change is respectively noted by arrows: KLF13 (↓↓), SULT4A1 (↓↑), NPDC1 (↓↑), S100A6 (↓↑), S100A2 (↓↑), NCOR2 (↑↓) RPTOR (↑↑), PPP1R18 (↑↑), UQCR11 (↑↓), PPP1CA (↑↓), CARNS1 (↑↓), and RPS6KB2 (↑↓). 

Although one environmental factor was chosen for each gene in this review, there is undoubtedly overlap in terms of a given factor having impact on more than one gene as one would suspect based on basic biological principles. TDO2 for example, was shown decades ago to be activated by the administration of THC to rats [213], and MCHR1 is not only co-localized with the dopamine synthesis marker, tyrosine hydroxylase but both may be regulated by the same transcription factor, HNF-1, suggesting an impact on MCHR1 from any environmental factor that might increase dopamine at the synapse. Furthermore, when considering gene products that interact, it should be kept in mind that epigenetic upregulation of one gene’s expression may lead to compensatory downregulation of a connected gene in a signaling cascade. Establishing the overall “tone” of the system is very difficult without actually measuring flux.

This review would not be complete without discussing what is known about the “flexible stem” hypothesis as it relates to epigenetics versus genome evolution [214,215,216]. Simply put, the theory is based on the premise that reversible changes in DNA precede changes that are subsequently hardwired by evolutionary adaptation in response to long-term shifts in the environment. The example presented by Muschick and colleagues [216] is the response of the species *Midas cichlids* when raised on diets either low or high in mechanical resistance. No differential survival was noted during the short-term course of the experiment, yet the changes they observed resembled the jaw bone differences seen between species of this genus. This phenotypic plasticity was considered by the authors to represent the flexible stem stage of evolution. From the perspective of molecular biology, this stage is entirely consistent with epigenetic effects. An analogy with psychotic disorders in humans may well be found in the season of birth effect for psychosis, which has been identified by some researchers to occur predominantly in those *without* a family history of psychosis [173]. Epigenetics likely underlies this phenomenon, and to the extent that MCHR1 haplotypes might be involved, a higher degree of methylation capability in the first exon would be most consistent with the flexible stem stage.

In conclusion, it is perhaps most remarkable that a study of genome-wide differential methylation caused by a particular environment known to be associated with psychosis, has revealed a relevant epigenetic signature in the gene TDO2, shown in several studies to be upregulated in psychosis. The promise of this type of research is that it will eventually enable a personalized record of environmental exposures relevant to disease, as well as providing tools to evaluate the efficacy of mitigation strategies.

## 6. Acronyms, Their Description and Any Associated Websites Used in This Review

3′ UTR—the untranslated region of mRNA that is 3′ (downstream) to the translation stop site; 5′ UTR—the untranslated region of mRNA that is 5′ (upstream) to the translation start site; 5hmC—a cytosine that has first been methylated to form 5mC and then the methyl group is hydroxylated in a separate enzymatic step, regarded as part of the methylation reversal process. 5mC—a cytosine that has been methylated at the 5th carbon in cytosine, ^m^C; aa—amino acid; AKT1—a serine/threonine phosphokinase (originally referred to as Protein Kinase B), leading to the phosphorylation of serine and threonine residues in specific proteins, important in signaling cascades; Alibaba2—a well-cited online program (cited by Kaushik et al. [216] Whitaker and Ostrander, [217]; Haro et al. [218]) that predicts transcription factor binding sites in a DNA sequence: http://gene-regulation.com/pub/programs/alibaba2/index.html (accessed on 1 November 2021); alpha-MSH—alpha-melanocyte-stimulating hormone, a member of the melanotropin family of peptides; ASSP—Alternative Splice Site Predictor program: http://wangcomputing.com/assp/ (accessed on 1 November 2021); Augustus—an online program that predicts open reading frames (ORFs): http://bioinf.uni-greifswald.de/augustus/submission.php (accessed on 1 November 2021); B3—the vitamins niacin and nicotinamide; BAT—brown adipose tissue, important in thermoregulation; CEBP—CCAAT/enhancer binding protein a transcription factor that enhances the function of GREs; CEU—a collection of DNA from individuals descended from a mixed population of northern Europeans in Utah; CpG—a dinucleotide pair of the bases cytosine and guanine in a DNA strand, frequently a site of cytosine methylation to yield ^m^C, i.e., 5mC; note that all CpG sites discussed in this review have associated SNPs, but unless specified otherwise in this review, all are too rare to be of relevance to psychosis; dbPTM database on protein post-translational modifications https://awi.cuhk.edu.cn/dbPTM/ (accessed on 1 November 2021); DRD2—one of the receptors for which dopamine is a ligand, in this case, the dopamine receptor-2; EMBOSS—an online tool hosted by the European Bioinformatics Institute https://www.ebi.ac.uk/Tools/seqstats/emboss_cpgplot/ (accessed on 1 November 2021); EWAS epigenome wide association study; FDR—*p* value that incorporates adjustment for the false discovery rate estimate; FEP—first episode psychosis patients; GRE—glucocorticoid response element, a nucleotide motif to which the glucocorticoid receptor protein binds; GSK3β—glycogen synthase kinase-3 β, a substrate for AKT1; when phosphorylated by AKT1-P, is inactivated; HCAR3—the hydroxycarboxylic acid receptor-3, responsive to niacin and carboxylic acids involved in lipid homeostasis in fat tissue; originally known as HM74; HM74—the human receptor originally identified as responsive to niacin; now also known as HCAR3; HNF-1—a transcription factor protein; among other regulatory functions, involved in affecting the expression of tyrosine hydroxylase; hnRNP-C—heterogeneous nuclear ribonucleoprotein C, a transcription factor that represses use of alternative 5′-cryptic acceptors; hnRNP-PTB—heterogeneous nuclear ribonucleoprotein- a transcription factor that represses use of alternative splicing 5′-cryptic acceptors, with the PTB designating its effector, polypyrimidine tract binding protein; IDO—indoleamine 2,3-dioxygenase, an enzyme that catalyzes the formation of kynurenine from tryptophan; iMethyl—website identifying location of cg probes http://imethyl.iwate-megabank.org (accessed on 1 November 2021); Ldpop—a website maintained by the National Cancer Institute that, among other things, enables determination of likely haplotypes for specific alleles: https://ldlink.nci.nih.gov/?tab=ldpop (accessed on 1 November 2021); MC5R—the melanocortin receptor-5, for which alpha-MSH is an agonist; MCH—melanin concentrating hormone, a member of the melanotropin family of peptides; MCHR1—melanin concentrating hormone receptor-1; MethBank—an online repository of differential methylation studies, searchable by gene name http://bigd.big.ac.cn/methbank (accessed on 1 November 2021); NCBI—the National Center for Biotechnology Information, https://www.ncbi.nlm.nih.gov (accessed on 1 November 2021); NCBI ORF finder—https://www.ncbi.nlm.nih.gov/orffinder/ (accessed on 1 November 2021); NF-kB—nuclear factor kappa-B subunit; modulates many genes, in response to many environmental stimuli, including stress and lack of sleep; ORF—open reading frame; PPI—pre-pulse inhibition, a measure of sensory gating employed in both studies of humans and animal models; PRS—polygenic risk score (for schizophrenia in this application); REM—random-eye-movement sleep, occurs late in the sleep cycle; thought to be important for memory consolidation, among other things; Scz schizophrenia; SNP—single nucleotide polymorphism, i.e., site for which there is more than one allele in the paired chromosomes containing a particular gene or intergenic region; note that all CpG sites discussed in this review have associated SNPs, but unless specified otherwise in this review, all are too rare to be of relevance to psychosis; TDO2—tryptophan 2,3-dioxygenase, an enzyme that catalyzes the formation of kynurenine from tryptophan; THC—the common acronym for Δ^9^-tetrahydrocannabinol, the psychoactive ingredient of cannabis; T-Rex™ HeLa cells are engineered to stably express a tetracycline repressor, lowering background expression for genes except those stimulated to be expressed; WAGGA—an online program that predicts coiled-coil regions in proteins, among other functions: https://waggawagga.motorprotein.de/ (accessed on 1 November 2021).

## Figures and Tables

**Figure 3 biomedicines-10-00561-f003:**
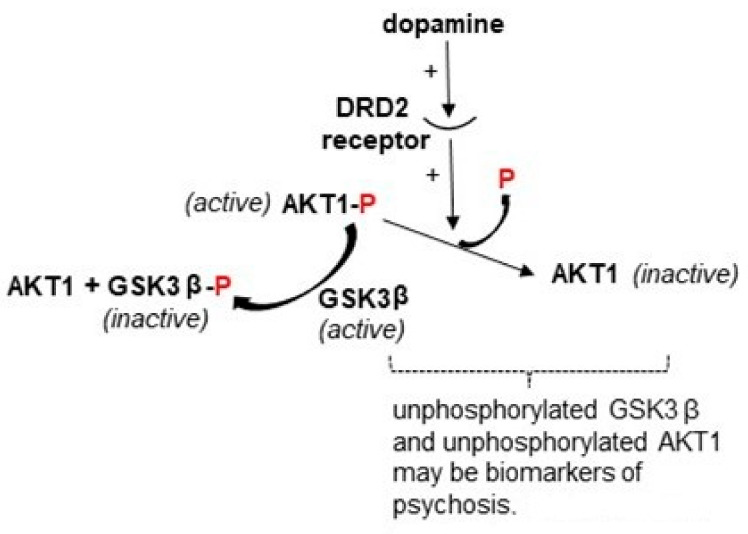
The dopamine signaling pathway that inactivates AKT1 and maintains activity of GSK3β by preventing its phosphorylation by AKT1.

**Figure 5 biomedicines-10-00561-f005:**
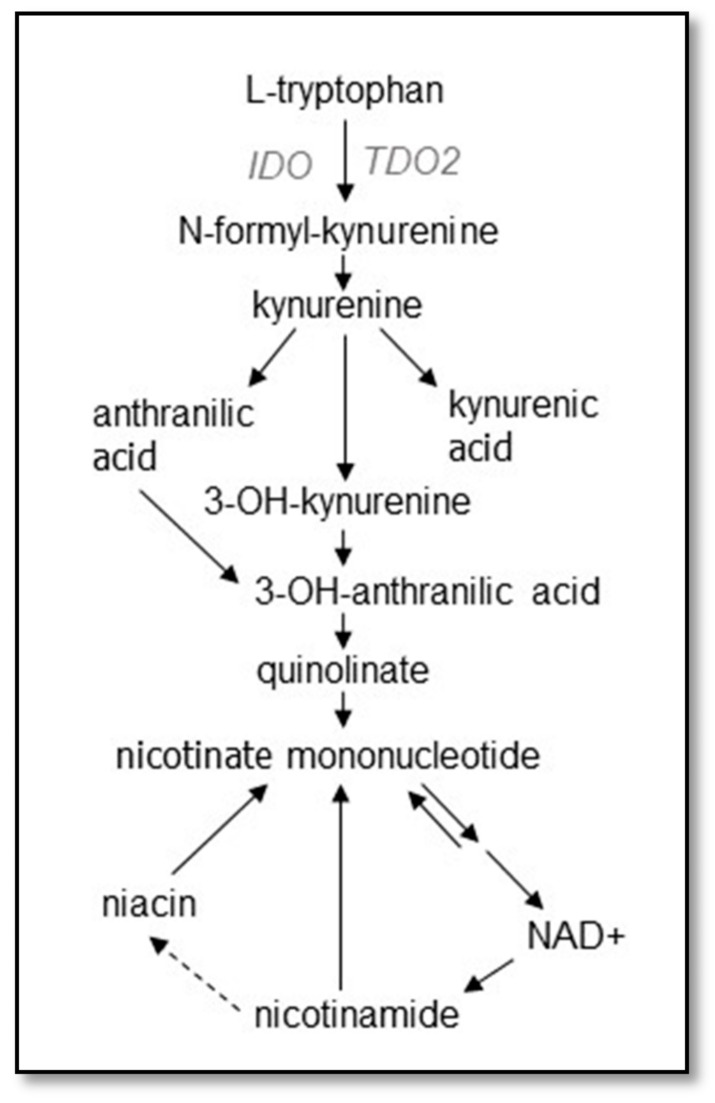
Primary components of the kynurenine pathway, from the substrate tryptophan to the end product, NAD+. The dashed line indicates the reaction is carried out by bacterial enzymes in the human gut.

**Table 1 biomedicines-10-00561-t001:** Summary of major points for the gene MCHR1.

**Important environmental cue** **reviewed**	Temperature, changes with season and latitude
**Direction of** **effect**	Cold temperature stimulates expression of peptide agonist; MCHR1 controls thermo-regulation to lower body temperature and conserve BAT
**Risk of schizophrenia (scz)** **consistent with known gene** **function?**	Yes: (1) season-of-birth effect (2) scz more common in populations in higher latitudes (3) ICV administration of peptide agonist in rats elicits auditory gating pattern consistent with scz (4) lower body temperature and over-dressing in scz (5) MCHR1 controls REM sleep, disrupted in scz (6) brain-region specific expression consistent with major pathways dysreg. in scz (7) co-localizes with tyrosine hydroxylase in the brain, a marker of dopamine synthesis
**Strength** **of genetic** **association**	Moderate: 4 positive studies; and 1 positive when part of a complex genotype; contributes to PRS; the alleles conferring risk depend on the population
**Epigenetic data** **of relevance?**	Yes: (1) 4 alleles linked in genetic association studies are located in (+/−) CpG sites (2) 1st exon CpG site is hypomethylated in psychosis (two studies) (3) evidence that methylation may control alternative splicing (4) smoking hypermethylates a CpG site in the 5′UTR region; predicted to affect expression of the 353aa MCHR1 protein reported by NCBI; hypermethylation in the promoter region may decrease expression of the 353aa protein and/or the alternative 422aa protein (the latter via alternative splicing). (5) the methylation difference between the rs133072 and rs13373 haplotypes is greatest in the age range of greatest risk for scz, 20 to 30 years of age
**Consistency of gene expression data**	One study: gene expression decreased in scz, but may be confounded by smoking based on methylation pattern identified in smokers

**Table 2 biomedicines-10-00561-t002:** Summary of major points for the gene AKT1.

**Important environmental** **cue reviewed**	Cannabis use
**Direction of** **effect**	Cannabis use *predicted* to inactivate AKT1 thru increased dopaminergic signaling, though animal data yields mixed results
**Risk of schizophrenia (scz)** **consistent with known gene** **function?**	Yes: schizophrenia is associated with increased dopaminergic tone, which should inactivate AKT1 via DRD2 agonists; drugs that are antagonists for DRD2 exert an antipsychotic effect
**Strength of** **genetic** **association**	Weak as a single locus: studies showing association as well as many showing lack of association; however, contributes to PRS
**Epigenetic** **data of** **relevance?**	Yes: (1) Methylation in exon-10 found to correlate with degree of brain activation in fear processing following THC; consistent with scz behaviors (2) Smoking leads to global hypomethylation of the gene; expected to incr. expr of AKT1 (3) Striking CpG islands in 5′UTR
**Consistency** **of gene** **expression** **data**	Mixed results for scz and related disorders Expression: Some studies show decreased gene expression, some no change and some increased expression; variability could be attributable to the effect of smoking seen in methylation data

**Table 3 biomedicines-10-00561-t003:** Summary of major points for the gene TDO2.

**Important environmental cue** **reviewed**	Stress
**Direction of** **effect**	Stress is *predicted* to increase expression of gene via the established effect of glucocorticoids to stimulate TDO2 mRNA expression
**Risk of schizophrenia (scz) consistent with known gene function?**	Yes: (1) periods of acute stress are often the trigger for psychotic breaks (2) upregulation of TDO2 metabolic product seen in several other disorders associated with psychosis
**Strength of** **genetic** **association**	Weak as a single locus: one study showing association only as part of complex genotype; however, contributes to PRS
**Epigenetic data** **of relevance?**	Yes: (1) hypomethylation of site in alt-exon-1 associated with childhood adversity; net effect cannot be predicted (2) hypermethylation of a site in intron 3 found in pregnant women engaged in night-shift work; function unclear (3) chronic mild stress in rats decreases methylation in promoter region (4) two GRE sites have CpG sites; methylation at these sites would incr. cortisol binding (5) hypermethylation of a CpG site in 5′UTR by maternal smoking; expected to downregulate expression; consistent with TDO2 expression results reported for smokers
**Consistency of** **gene expression data**	Consistent: (1) mRNA upregulated in scz (4 studies) and bipolar disorder with psychosis (2 studies) (2) expression found to be decreased in smokers, consistent with epigenetic data for smokers and with two expression studies of smokers in controls, scz and BPD wi psychosis; but not enough of an effect to overpower upregulation by the disorders

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
