# Peer review of "The Epigenetics of Psychosis: A Structured Review with Representative Loci"

_biomedicines, 2022, doi:10.3390/biomedicines10030561_

Round 1

Reviewer 1 Report

This review describes the involvement of three genes (MCHR1, AKT1 and TOD2) in psychosis and schizophrenia and how some environmental factors can affect their expression through epigenetic mechanisms. The manuscript is well written and suitable for publication.

A presence of male bias in schizophrenia has been shown, with males showing higher incidence and lower age of onset than females. Can the author comment about sex differences, if any, in the described genetic and epigenetic mechanisms regulating the expression of examined genes?

Author Response

The author is grateful for the time and effort the reviewer invested in reviewing this paper. The supportive comments are much appreciated and an important point is raised.

1) Specific comment from reviewer: What regulatory or differential epigenetic patterns may underlie the increased risk for schizophrenia in males? 

Response: This is a key point that was not addressed in the manuscript. Although no gender-specific methylation differences have been reported for the three genes reviewed in the paper, regulation by estrogen is reported for the enzymatic activity of TDO2, and data suggests that TDO2 expression may be under negative control by estrogen in breast cancer, likely via indirect interactions. There are as yet no epigenetic data concerning estrogenic control of TDO2 in this regard, though it has been identified (also in breast cancer) for the other initiating enzyme of the kynurenine pathway, IDO. To cover this point, the following text has been added to the manuscript, p. 18, 2nd paragraph.

“There is very likely a role for sex hormones in the environmental modulation of psychosis, as males are at greater risk for this disorder [179], and the risk for psychotic disorders increases in the peri- and post-menopausal time [180]. Of the three genes in this review, it may be TDO2 regulation that illustrates some degree of relevant gender-specificity.  Badawy [181] reports that estrogen directly inhibits the activity of the enzyme by affecting its ability to bind heme, although this is a finding not yet confirmed by other groups. In addition, studies show upregulation of TDO2 expression seen in estrogen-receptor-negative but not in estrogen-receptor-positive breast cancer [182], suggesting that expression of TDO2 may be under negative regulatory control by estrogen in this setting.”

Reviewer 2 Report

This is a comprehensive review focusing on three genes, melanin concentrating hormone receptor-1 (MCHR1), serine-threonine protein kinase AKT1 and tryptophan 2,3-dioxygenase (TDO2), and their association with psychotic disorders through differential methylation. The author thoroughly reviews the topic and raises some interesting questions.

Specific comments:

The Tables are very informative and succinctly systematize a lot of data.

Some sections are very extensive with perhaps some unnecessary details. That makes them difficult to follow at times which affects the smooth reading of the whole text. Some examples include:

Lines 78-105: the whole paragraph may be interesting, but not really necessary with all the methylation details.

Lines 153-160: the weight control part can be shortened or omitted all together.

Lines 210-216: the experiments in question do not need to be described in such detail. The results presented in lines 216-221 are more than enough; one can always turn to the cited original research for more details.

Lines 377-484: the AKT1 review starts at line 377, but it’s not until line 484 that the methylation is being mentioned; consider shortening.

The manuscript could be improved if a short conclusion or summary is added following the discussion.

Minor points:

There is an extra full stop in line 372.

“NAD” should be corrected to “NAD+” throughout the text (e.g. line 574).

Author Response

The author is grateful for the time and effort the reviewer invested in reviewing this paper. The suggest edits are very much appreciated and will improve the manuscript, particularly the condensing of some concepts into more concise paragraphs.

1) Specific comment from reviewer: Lines 78-105: the whole paragraph may be interesting, but not really necessary with all the methylation details

Response: Yes, good point. Original lines 78-105 (currently on page 2, last paragraph) have been condensed by 60%, to eliminate unnecessary detail, while still retaining the concept of methylation reversal leading to the intermediate 5hmC, which becomes important later on for understanding splicing data on MCHR1.  This section now reads:

“To set the stage for what is to come, it is important to point out a great paradox of methylation, in that 5mC renders the cytosine base more susceptible to transitioning to thymine (T) through spontaneous deamination, the probability of which obviously increases with time [14]. In an environment that promotes consistent epigenetic methylation of a gene, the end result can be the loss of methylation capability at that site, which should be maladaptive given the existing environment that encouraged it in the first place. The explanation for this apparent conflict may lie in the enzymatic reversal of methylation which involve conversion of 5mC to 5-hydroxy-methyl-cytosine (5hmC) as the first step, followed by oxidation intermediates and base excision repair [15]. In such a way, the epigenetic control of the site can be maintained and the replacement of a C by a T minimized.”

2) Specific comment from the reviewer: Lines 153-160: the weight control part can be shortened or omitted all together.

Response: The original lines 153-160 have been eliminated from that paragraph, while condensing the text by ~49% for integration with the following paragraph.  Part of the problem with this section was likely the lack of cohesive flow between related points, and hopefully these edits have corrected the issue. This section (currently p. 4, 2nd paragraph) now reads:

“Whereas the alpha-MSH melanotropin system is closely tied to seasonal photoperiod responses in Siberian hamsters and other rodents [29], no response to photoperiod can be detected in the MCH system in those species, and there is only one report of a possible photoperiod effect of long days to depress MCH expression in higher animals, i.e. in sheep [30]. Rather, the primary seasonal cue for activating the MCH system may be temperature, as even short-term exposure to cold can increase the expression of the mRNA for MCH peptide [31]. Increased MCH function in response to colder environmental temperatures could be tied to conservation of energy stores in winter months, based on the involvement of mammalian MCH in creating and preserving fat stores [32,33], and the strong seasonal trend of this effect [34]. Thermogenesis from fat occurs primarily from brown adipose tissue (BAT), including in humans [35]; expression of MCHR1 has been demonstrated in BAT [36]; and Izawa et al. [37] have shown that activation of MCH producing neurons negatively regulates energy expenditure in murine BAT tissue.  Consistent with this theme, prior work demonstrated chronic infusion of MCH in wildtype mice causes not only a decrease in energy expenditure, but also a lower body temperature [38]. These findings could be of particular relevance to the altered thermoregulation by individuals with schizophrenia, and the fact that they frequently overdress, even in summer [39,40]. A study of patients both on and off neuroleptics illustrated significantly lower body temperature in cases as compared to controls [41], and basal metabolic rate is apparently altered as well [42].  This characteristic of the disease appears to have long-predated the advent of neuroleptic use [43]. “

3) Specific comment from the reviewer: Lines 210-216: the experiments in question do not need to be described in such detail. The results presented in lines 216-221 are more than enough; one can always turn to the cited original research for more details.

Response: Yes, good point.  The experimental details have been omitted and the section now reads (currently page 5, 1st paragraph):

“Thus, an investigation of intra-cerebroventricular administration of MCH and alpha-MSH was undertaken in an animal model of the disrupted auditory sensory gating phenotype observed in individuals with schizophrenia. The results demonstrated the functional antagonism between the peptides first observed in teleost fish [25-27] extends to auditory gating, with MCH disrupting auditory gating in dose-responsive manner and eliciting a pattern observed in schizophrenia subjects, whereas alpha-MSH enhanced auditory gating beyond what was typically observed in controls [50].”

4) Specific comment from the reviewer: Lines 377-484: the AKT1 review starts at line 377, but it’s not until line 484 that the methylation is being mentioned; consider shortening.

Response: Yes, portions of this section were not very relevant and/or convoluted because of the lack of consistent findings.  Thus, despite the fact that the AKT1 section was already somewhat smaller than those for the other genes, the length has been reduced ~25% by removing the following text (now missing between from pages 10 and  11).  The associated text was also removed from Table 2.

  1. a) This section is deemed unnecessary because the inconsistencies between auditory gating and PPI results have already been discussed: “Yet, studies of the effect of THC on the other sensory gating trait documented for psychosis, PPI, have yielded mixed results, with Nagai et al. (2006) reporting diminished PPI following acute administration of THC, in conjunction with a decrease in the startle response (minus the prepulse) in a murine model, yet Long et al. (2010) and Todd et al. (2020) reporting that THC acutely facilitates PPI in conjunction with a decrease in startle response (minus the prepulse) in mice. In a review by Eric Miller (2021), the lack of reproducibility of this test has been shown to be mediated by numerous confounding factors in animal models, including strain, age, sex, reproductive, species, habituation, socialization and the baseline startle response.”
  2. b) This section is deemed unnecessary because it pertains to post-translational modification, which is not relevant to expression changes that might be attributable to epigenetic effects on AKT1:

“Research into the phosphorylation of the kinase and its substrate, GSK3β (Figure 3) has also yielded mixed results.  In human studies, GSK3β phosphorylation has been reported to be increased in drug-naïve, first episode psychosis (FEP) patients (Joaquim et al., 2018), whereas Ferriera et al. (2014) report that GSK3β phosphorylation decreased in FEP subjects when unmedicated at baseline, though admittedly only known to be medication-free for 2 weeks.  Treatment of those same patients with olanzapine increased phosphorylation.  In rats, Seillier et al. (2020) reported a decrease in AKT1 phosphorylation at the highest dose of THC tested (1 mg/kg), and a decrease in GSK3-β phosphorylation at all doses tested (0.1 mg/kg to 1.0 mg/kg), consistent with an increase in the dopaminergic signaling cascade depicted in Figure 3.  In another study, treatment of rats for 14 days with cannabinoid receptor-1 (CB1) agonists similar to THC, resulted in decreased phosphorylation of AKT1 (Papadogkonaki et al., 2019); however, a different picture emerges with chronic THC administration for 30 days in mice: phosphorylation of AKT1 was increased (Ibarra-Lecue et al., 2018). Of note, Renard et al. (2017) found that the dephosphorylation effects on AKT1 in rats were specific to the age period equivalent to adolescence in humans.”

5) Specific comment from the reviewer: The manuscript could be improved if a short conclusion or summary is added following the discussion.

   Response: Yes, good point. The best way to come up with a concise concluding comment in regards to the extensive material covered is to focus on the most remarkable result in my view:

“In conclusion, it is perhaps most remarkable that a study of genome-wide differential methylation caused by a particular environment known to be associated with psychosis, has revealed a relevant epigenetic signature in the gene TDO2, shown in several studies to be upregulated in psychosis. The promise of this type of research is that it will eventually enable a personalized record of environmental exposures relevant to disease, as well as providing tools to evaluate the efficacy of mitigation strategies.”

6) Specific comment from the reviewer: There is an extra full stop in line 372.

   Response: Thank you, it is now corrected.

7) Specific comment from the reviewer: “NAD” should be corrected to “NAD+” throughout the text

   Response: Thank you, this mistake is now corrected.